# Dual-Square-Split-Ring-Enclosed Microstrip-Based Sensor for Noninvasive Label-Free Detection

**DOI:** 10.3390/ma15217688

**Published:** 2022-11-01

**Authors:** Air Mohammad Siddiky, Mohammad Rashed Iqbal Faruque, Mohammad Tariqul Islam, Sabirin Abdullah, Mayeen Uddin Khandaker, K. S. Al-Mugren

**Affiliations:** 1Space Science Centre (ANGKASA), Institute of Climate Change (IPI), Universiti Kebangsaan Malaysia, Bangi 43600, Malaysia; 2Centre for Advanced Electronic and Communication Engineering, Department of Electrical, Electronic & Systems Engineering, Universiti Kebangsaan Malaysia, Bangi 43600, Malaysia; 3Centre for Applied Physics and Radiation Technologies, School of Engineering and Technology, Sunway University, Bandar Sunway 47500, Malaysia; 4Department of Physics, College of Science, Princess Nourah Bint AbdulRahman University, Riyadh 11671, Saudi Arabia

**Keywords:** dual square rings, high compact ratio, metamaterial, microstrip sensor

## Abstract

In this article, we present the use of a metamaterial-incorporated microwave-based sensor with a single port network for material characterization. The proposed sensor consists of a microstrip patch layer enclosed with a dual-square-shaped metamaterial split-ring. This structure has the dimensions of 20 × 20 × 1.524 mm^3^ and a copper metallic layer is placed on a Rogers RT 6002 with a partial back layer as a ground. Two resonant frequencies are exhibited for applied electromagnetic interaction using a transmission line. The dual split rings increase the compactness and accumulation of the electromagnetic field on the surface of the conducting layer to improve the sensitivity of the sensor. The numerical studies are carried out using a CST high-frequency microwave simulator. The validation of the proposed sensor is performed with an equivalent circuit model in ADS and numerical high-frequency simulator HFSS. The material under test placed on the proposed sensor shows good agreement with the frequency deviation for different permittivity variations. Different substrates are analyzed as a host medium of the sensor for parametric analysis.

## 1. Introduction

Microwave sensors are widely utilized in many industries such as agriculture, the biomedical sector, and electronics because they offer a short reaction time, a wide sensing range, excellent precision, no temperature influence, and are operative in harsh environments [1,2]. Due to unique metamaterial features [3,4,5] with negative effective parameters, microwave sensors show unique properties through the manipulation of the electromagnetic wave. The properties of the electromagnetic wave can be varied by the geometric engineering of the metallic layer to influence the interaction of the applied signal, which has recently attained a high degree of interest in a widespread manner [6]. Different types of metamaterial unit cells [7] have been utilized for different applications in microwave regions to enhance the accumulation of electromagnetic fields, and the impact of this artificial technology has introduced novel features that are not possible to achieve in natural materials. 

Electromagnetic waves have generated novel scattering properties to control the higher-frequency spectrum for wireless communications in the last two decades [8]. For material characterization, complementary split-ring [9]-based sensors were utilized to enhance the sensing performance, which was printed on a FR-4 substrate. Different types of microwave sensors have been developed with different methodologies for sensing applications within the operating frequency range. For liquid characterization [10], a microstrip-fed rectangular patch antenna using a Rogers RO 3035 as a host medium with two port networks was presented where the capillary glass tube lay parallel to the surface of the sensor, where the large diversity of geometric orientation [11] of the chiral structure showed a new way to manufacture three-dimensional structures. The other methodology using metamaterial absorber technology was utilized to accumulate the electromagnetic signal with a silver-based metallic layer to form two concentric rings [12], which produced two narrow absorption peaks. The metamaterial-unit-cell-based sensors [13] were utilized for blood glucose concentration and showed higher sensitivity with an increasing number of unit cells for different dielectric properties. For chemical compound characterization [14], a square split-ring resonator was deployed for a microstrip-based sensor, which was squeezed to the narrow width of the slot inside a cylindrical tube and utilized a microwave frequency in the range of 1.0–1.2 GHz with different sample concentrations. To determine the amount of methanol in methanol-contaminated local spirit samples, a square split-ring resonator metamaterial-based transmission line sensor [15] was designed with a 150 MHz bandwidth. A metamaterial-integrated rectenna system [16] was developed to characterize material properties where four metamaterial unit cells were constructed as a first layer, a patch antenna on the second layer, and a rectifier circuit on the third layer of the sensor. 

In this research work, we introduced a metamaterial-enclosed-microstrip-based sensor where this design was compact in size for the incorporation of the dual-square split rings. This resonant structure as well as the hosting transmission line had an impact on the performance of the planar microwave sensor. The dual-split-rings-enclosed microstrip-based sensor with the dimensions of 20 × 20 mm^2^ produced dual resonant frequencies at 3.56 and 4.14 GHz. The sensitivity and R-square were 210 and 0.9943, respectively, where the proposed design showed a good linear relationship with the resonant frequency for different dielectric constants. Moreover, the compactness ratio of the proposed sensor was 3.57.

## 2. Metamaterial-Based Sensor Configuration

The magneto-dielectric substrate layer, which is put beneath the patch, is made up of two square split rings that enclose the patch. This strategy shifts the resonant frequency to the lower-frequency region to miniaturize the design. The square-shaped split rings marked as a white color and microstrip patch as a brown color in Figure 1a are printed on the substrate Roger RT 6002, with a thickness of 1.524 mm, and it is made up of three layers of varying thicknesses, where the copper has a thickness of 0.035 mm. The upper and lower layers consist of copper material that acts as a conducting channel for the induced current circulation, as shown in Figure 1b. The substrate acts as a host medium in Figure 1c, which is placed in the middle layer and is used to confine the polarization impact due to the fringing effect of the applied electromagnetic signal. The parametric values are shown in Table 1. The patch has the dimensions of 12 × 12 mm^2^ with an inner slot that has a width of 0.5 mm. A single-port-based transmission line with a 1 mm feed width and a 5 mm length is connected to the body of the microstrip patch. 

In the back, the dielectric substrate is covered with the partial ground layer, which has the dimensions of 20 × 6.5 mm^2^. The length of the outer split ring is 16 mm and the length of the inner split ring is 14 mm. 

The metamaterial unit cell adds the inductive effect for the metallic arm, where the gap between the metal is exhibited the shunt capacitive effect with the dielectric and series effect between the metallic rings. This mutual impedance effect shifts the resonant frequency to the lower frequency and increases the compactness of the design. The equivalent LC circuit effect produces the resonant frequency. A split-ring resonator is magnetically connected to transmission lines such as Microstrip or coplanar waveguides, where the split-ring resonator is electrically coupled to these transmission lines. In an LC resonator, a square ring works as an inductor and the spaces between rings and their neighboring ground plane operate as capacitors [17]. The metallic strips are located on the substrate, where S is the distance between two parallel plates and W is the conductor width. The effective dielectric constant is $\varepsilon_{e}$. The inductance value may be adjusted for many aspects such as length, turn ratio, and breadth of the metallic strip. The capacitance effect between the metal and the dielectric substrate, two parallel plates, and the fringing field is also taken into consideration. a is the diameter of the strip, h is the thickness of the host medium, and t is the thickness of the metallic layer.

The inductance of the metallic strip can be expressed as:(1)$$L=2\;\times \;10^{-2}\;l\;\left[\ln\left(\frac{a}{w+t}\right)+1.193+0.2235\;\frac{w+t}{l}\right]$$

The capacitance of the structure can be expressed as:(2)$$C\;(\mathrm{pF})=\frac{\varepsilon_{e}10^{-3}\;K\left(k\right)\;}{18\pi \;K^{\prime }\left(k\right)}\;(N-1)\;l$$
where *K* =$\;tan^{2}(\frac{a\pi }{4b}$); a = $\frac{w}{2};\;b=\frac{w+s}{2}$.

For this suggested sensor, the equivalent circuit model is drawn in Figure 2. The inductance L1 and capacitance C1 are taken for the transmission line with the host medium. The mutual impedance effect for split rings with the patch is presented as an LC tank circuit with the series inductance effect, which is separated into two branches as (L2, L4, C2) and (L3, L5, C3). The capacitive effect with the patch of the host medium is presented as L6, C4.

Different geometric shapes in the metallic layer on the host medium, which is excited by electromagnetic frequency, produce different impedance responses for the applied frequency and show a complex response for the high-frequency spectrum. For this proposed sensor, we first use a microstrip patch on the substrate in Figure 3a and produce the resonant frequency in the X band region, which is shown in Figure 3b. Adding a single split ring that encloses the microstrip patch produces resonant frequencies at 4.14 GHz, which is shifted to a lower-frequency region. In design 3, dual split-ring resonators with a microstrip patch produce dual resonant frequencies at 3.56 and 4.14 GHz. The metamaterial-unit-cell-incorporated microstrip patch where dual-square split rings enclose the microstrip patch is selected for the suggested sensor. The combination of the split rings with the microstrip patch enhances the accumulation of the applied electromagnetic spectrum. Moreover, the introduction of dual split rings to the microstrip-based sensor increases the compactness of the design. 

## 3. Methodology 

### 3.1. Metamaterial Unit Cell

The artificially structured metamaterials can control the magnetic and electric response over a broad frequency range, whereas the tailored geometric layer can change the response of EM radiation. The scattering properties for the metal–insulator-based artificial structure are related to the mutual impedance value of the medium. Figure 4 shows that ZY-plane walls are perfect electric conductors (PECs) under certain boundary conditions, whereas XY-plane walls are ideal magnetic conductors (PMCs). For numerical calculation of the scattering properties for an applied electromagnetic wave, a CST-based high-frequency structural simulation is utilized within the specific frequency range. The finite element method is performed for the meshing techniques of the proposed design. The split-ring resonator (SRR) is utilized to reduce the size of the antenna. According to this, a magnetic field perpendicular to the ring (loop) creates a current in the ring, which causes magneto-dipoles to form. As a result, the substrate permeability and permittivity product will rise. This approach may compensate for bandwidth deterioration, allowing for antenna miniaturization. 

### 3.2. Metamaterial-Based Sensor

Building a microstrip structure with metamaterials can be used to design a small structure. If the metamaterials utilized have high permeability and operate as a magneto-dielectric (MD) substrate, this does not impair performance efficiency. This leads to a change in the total impedance value with the complex response and introduces the polarizability state to the lower-frequency region. The use of metamaterials in the sensor design may result in size reduction, enhanced gain, and increased bandwidth. Metamaterials can be employed for a variety of roles in the sensor design depending on the technical requirements. This proposed sensor in Figure 5 is a single-transmission-line-based microstrip sensor with a partial ground structure for dielectric material characterization.

The width of the microstrip can be [18] calculated as: (3)$$W=\frac{1}{2f_{r}\sqrt{\varepsilon_{0}\mu_{0}}}\sqrt{\frac{2}{\varepsilon_{r}+1}}$$
where C is the free space velocity of light and *ε_r_* is the dielectric constant of the substrate.

The effective dielectric constant of the rectangular microstrip patch antenna can be evaluated as:(4)$$\varepsilon_{ff}=\frac{\varepsilon_{r}+1}{2}+\frac{\varepsilon_{r}-1}{2}\;\frac{1}{\sqrt{1+\frac{12h}{w}}}$$

The actual length of the patch (*L*) of the microstrip can be calculated as: (5)$$L=L_{eff}-2\Delta L$$
where $L_{eff}=\frac{C}{2f_{r}\sqrt{\varepsilon_{eff}}}$.

The length extension can be expressed as:(6)$$\frac{\Delta L}{h}=0.412\frac{\left(\varepsilon_{ff}+0.3\right)\left(\frac{W}{h}+0.264\right)}{\left(\varepsilon_{ff}-0.258\right)\;\left(\frac{W}{h}+0.8\right)}$$

### 3.3. Effective Parameter

A variation in cell dimensions can satisfy the resonant frequency requirement. It should be emphasized that for a metamaterial to meet the criteria of homogeneity, the cell size must be substantially smaller than the guided wavelength. According to reports, numerical simulation of the unit cells is unable to produce the desired results. As a consequence, the cell sizes are tweaked until the simulation results confirm the basics of the structure. The size of unit cells generating acceptable results is determined using an optimization computational approach to obtain sufficient output in less time. According to Equations (7) and (8), the resonance frequency and thickness of the metamaterial slab are inversely proportional to the effective permeability and permittivity. As a result, the resonance frequency may be readily moved to lower frequencies by increasing the product of effective permeability and permittivity. The effective permittivity and permeability [19] can be calculated as:(7)$$\mathrm{Effective}\;\mathrm{permittivity}\;\;\varepsilon_{r}=\frac{2}{j\pi fd}\times \frac{\left(1-S_{21}-S_{11}\right)}{\left(1+S_{21}+S_{11}\right)}$$
(8)$$\mathrm{Effective}\;\mathrm{permeability}\;\mu_{r}=\frac{2}{j\pi fd}\times \frac{\left(1-S_{21}+S_{11}\right)}{\left(1+S_{21}-S_{11}\right)}$$

## 4. Results and Discussion

### 4.1. Results of Effective Properties for the Metamaterial Unit Cell

In Figure 6a, the proposed metamaterial unit cell exhibits resonant frequencies for the transmission coefficient at 1.74 and 2.15 GHz and for the reflection coefficient at 1.88 GHz. In Figure 6b,c, the effective parameters such as effective permittivity and effective permeability are presented for the metamaterial unit cell. The real negative permittivity value is shown from 1.77 to 1.81 and 2.15 to 2.44 GHz and the real negative permeability is shown from 1.69 to 1.79 and 2.06 to 2.17 GHz. The negative region of the effective parameters enhances the confinement of the electromagnetic wave on the structure and produces negative electric susceptibility where the electric flux density follows the opposite direction of the applied electric field. This effect increases the accumulation of the applied electromagnetic field and the displacement current on the conducting surface. 

### 4.2. Electric and Magnetic Field, Surface Current, and Power Flow Distribution

At 3.56 GHz, the electric field distribution in Figure 7a exhibits a strong accumulation of the electric field on the upper and lower parts of the square split-ring resonator, where the magnetic field in Figure 7b is induced on the outer metallic ring vertically to resonate the circuit. The confinement of the electric and magnetic field in the microstrip-based design produces the circulating current on the surface, which is distributed on the outer metallic strip of the design, which is shown in Figure 7c. For the second resonant frequency, the electric field is distributed as the first resonant frequency due to the near-range wavelength as the second resonant frequency. By altering the magnetic field, the time-varying electric field component is created. The magneto-static field is strengthened by the artificial magnetic component that propagates perpendicular to an electric field. Magnetic fields that alter over time affect a substance’s magnetic dipole moment and magnetization force. The surface current is distributed on the inner-square split ring adjacent to the microstrip patch. The imposed electric field component changes the polarizability state, where the magnetic field affects the dipole moment of an atom and enhances the magnetic responses. In this proposed design, a strong sensitivity part is shown on the adjacent area of the patch with the outer split rings.

The power flow distribution is presented in Figure 8 for the proposed microstrip-based sensor. An aerial view of the power flow distribution is shown in Figure 8a and a lateral view is shown in Figure 8b. A 50 Ω based microstrip line is connected to the sensor and power is distributed from the single port, which provides a quasi-TEM wave to the design. For the first resonant frequency at 3.56 GHz, the power is distributed strongly at the edge of the metallic ring and the gap between the two outer split rings. At the second resonant frequency, the power is strongly accumulated at the edge between the microstrip patch and the inner split ring, which is adjacent to the patch. The lateral view shows that the power is radiated around the outer split ring at 3.56 GHz and the outer split ring at 4.14 GHz. 

### 4.3. Radiation and Dispersive Properties

Integration of unit cells with radiated components is a simple method that allows them to behave as insulators by reflecting surface waves due to their negative effective response. The obtained gain is dependent on the number of unit cells and the resonance frequency. The use of metamaterial superstrates in the antenna design enhances the gain while simultaneously increasing the size of the sensor and its thickness. In Table 2, the radiation efficiency, total radiation efficiency, gain, realized gain, and radiation plots are observed for four resonant frequencies at 3.56, 4.14, 7.12, and 8.10 GHz. The radiation efficiency and total radiation efficiency show near values, which express the low losses of the proposed microstrip-based structure. Due to low loss, the gain and realized gain exhibit less difference with a noticeable radiation plot for the first two resonant frequencies. The dispersive property for Rogers RT 6002 as a host medium is presented in Figure 9.

### 4.4. Simulated and Measured Result

The metamaterial-based multiband dipole microstrip sensor with split loops functions as a near-field resonant parasitic component. The number of operational frequencies is determined by the size of the inductive loop and the capacitive effect of the host medium. The split-ring resonator induces a circulating current that establishes the energy across the gap and produces the resonance in the different frequency region, where the proposed metamaterial-based sensor exhibits the resonant frequencies at 3.56 and 4.14 GHz. This suggested microstrip-based sensor is validated by CST microwave studio, high-frequency structural simulator (HFSS), and the equivalent circuit model in ADS, and the corresponding results are shown in Figure 10. The numerical results vary due to the port specification, boundary condition, data acquisition method, and using manual data for the equivalent circuit model. For experimental purposes, the fabricated prototype using Rogers RT 6002 as the host medium of the sensor is connected with a microstrip line through a 50 Ω based coaxial cable, and the corresponding results show discrepancies due to the fringing field effect on the proposed sensor. The corresponding results of the reflection coefficient are shown in Figure 11. The frequency deviations in the reflection coefficient for experimental and numerical results are shown for different reasons: the microstrip design is attached to the cables where the received wave might be out of phase, resulting in an error, undesired electrical disturbance, drift error, etc.

## 5. Parametric Analysis

Different dielectric substrates are utilized to observe the scattering properties such as reflection coefficient for Rogers RT 6002, FR-4, Rogers RO 4003, Rogers RT 5880, and polyimide, where the corresponding results are shown in Figure 12. The FR-4 substrate as the host medium exhibits two resonant frequencies at 3.11 and 3.6 GHz with depth notches at −9.4 and −32, respectively, which are shown in Table 3. The higher dielectric constant is responsible for the variation in polarization state for the applied electric field component. The RT 5880 produces the resonant frequencies at 3.94, 4.6, 7.89, and 8.98 GHz with depth notches at −11.5, −15.3, −12.84, and −26.41, respectively; RO 4003 generates the resonant frequencies at 3.33 and 3.84 GHz with depth notches at −22.84 and −24.24, respectively. Decreasing the permittivity values of the dielectric substrate as the host medium, the resonant frequencies shift to the higher-frequency region. For the polyimide substrate, the resonant frequencies are located at 3.4, 3.88, 6.67, and 7.58 GHz with depth notches at −26.32, −25.10, −8.28, and −12.89, respectively. Rogers RT 6002, which is utilized in the proposed sensor, produces resonant frequencies at 3.56, 4.14, 7.12, and 8.10 GHz with depth notches at −19.8, −23, −11, and −21, respectively. This substrate with the suggested design configuration shows a more stable depth notch for four resonant frequencies.

## 6. Microstrip-Based Sensor with MUT

The four resonant frequencies of the proposed sensor are observed for different MUTs, where the MUTs are placed on the metallic layer of the sensor structure. In Figure 13, the volumetric dimension is shown for different MJTs for the applied electromagnetic signal via a single-port-based microstrip sensor. With the increase in the dielectric constant, the effective permittivity is changed due to the variation in the capacitance effect. For MUT 1 (ε_r_ =1), the resonant frequencies in Figure 14 are located at 3.44, 3.91, 6.69, and 7.69 GHz with depth notches at −29.6, −24.9, −13.9, and −15.84, respectively. Increasing the dielectric constant to 1.5 for MUT 2, the resonant frequencies shift to the lower-frequency region at 3.29, 3.74, 6.58, and 7.39 with depth notches at −23.8, −16.13, −15.7, and −10.35, respectively. 

With the increase in the dielectric constant for different MUTs, the resonant frequencies are distributed at 3.14, 3.58, 6.30, and 7.10 GHz with depth notches at −36.2, −11.5, −17.22, and −7.37 for MUT 3, respectively (ε_r_ = 2.5); at 3.03, 3.46, 6.06, and 6.85 GHz with depth notches at −34.34, −8.83, −15.65, and −5.9 for MUT 4, respectively (ε_r_ = 3); at 2.92, 3.32, 5.84, 6.60 GHz with depth notches at −17.85, −7.21, −14, and −4.7 for MUT 5, respectively (ε_r_ = 3.5); at 2.83, 3.22, 5.65, and 6.38 GHz with depth notches −21, −6, −11.4, and −4 for MUT 6, respectively (ε_r_ = 4). Different samples with the variation in the dielectric constant are placed on the sensor to observe the sensitivity analysis. The compactness of the design is increased due to the incorporation of a metamaterial unit cell, and a higher accumulation of the electric and magnetic field is shown in dual square rings adjacent to the microstrip patch.

The sensitivity performance is evaluated using Equation (9), where $f_{1}^{\prime }$ is denoted for the first resonant frequency and $f_{2}^{\prime }$ is denoted for the second resonant frequency of the proposed sensor with the MUT. The sensitivity of the sensor and resolution are the two most significant characteristics. Sensitivity is a physical property that defines the response of the sensor with the variation in resonant frequency within the operating frequency range. For this proposed sensor, the sensitivity performance evaluation is modified for dual resonant frequencies for different MUTs to observe the frequency deviation within the operating frequency range. For sensor performance evaluation, the sensitivity can be expressed as:(9)$$S=\left(f_{2}^{\prime }-f_{1}^{\prime }\right)\;\left(\varepsilon_{r}-1\right)\times 1000\;(\mathrm{MHz})$$

For *ε_r_* = 1.5, the scattering properties are changed due to the introduction of the MUT on the proposed sensor. The fringing effect is distributed to the MUT, which changes the electric polarization for the applied high-frequency spectrum. The resonant frequencies shift to the higher-frequency region and a high-frequency deviation is shown at the first and second resonant frequencies with a sensitivity of 235, as mentioned in Table 4. An increment in dielectric constant to 2 leads to a greater frequency deviation to the lower-frequency region at 3.29 and 3.79 GHz due to the variation in the capacitance effect for the sample on the sensor, where the sensitivity for the two frequencies is 450. As a consequence, the rest of the samples with variations in the dielectric constant to 4 show the following sensitivity performances for the two resonant frequencies: 660 for *ε_r_* = 2.5, 860 for *ε_r_* = 3, 1000 for *ε_r_* = 3.5, and 1170 for *ε_r_* = 4. The sensitivity performance for different dielectric constants is observed on the proposed sensor where the depth notch variation can be ignored to take into account the deviation in the dual resonant frequencies. 

For the first resonant frequency in Table 5, the increment in the dielectric constants is shown in the steady linear relation for different sample materials, where the linear equation is y = 372.86x − 296.19 and the sensitivity is R^2^ = 0.9943. For different dielectric constants, the proposed sensor exhibits good agreement with linear sensitivity performance. The frequency deviation is related to the change in the dielectric constant of the MUT, which is adjacent to the sensor. However, the resonant frequencies at higher-frequency regions with the change in permittivity value of the sample materials show the linear relation with the change in the mutual impedance for the transmission line feed electromagnetic signal of the proposed sensor. The linear relationship with the sensitivity performance with the variation in sample material properties introduces the electric polarization through the MUT, and the resonances are shifted due to the change in permittivity values. Moreover, the proposed sensor can be applied with different depth notches due to the sensing evaluation approach using the frequency deviation with the change in dielectric constant values. The resolution of the sensor performance with the change in different MUTs shows a noticeable response and the fractional dielectric constant for different MUTs can be detected within the operating frequency range. The measured result with MUTs is presented in Figure 15 where the FR-4 substrate is used as the MUT on the proposed sensor for noninvasive label-free detection.

The recently reported high-performance metamaterial-inspired sensing applications are based on different kinds of metamaterials. The derived high-quality-factor metamaterial units are discussed in Table 6 for high-precision sensing applications, in terms of the sensitivity and resolution to find some new design routes for sensing applications. The abovementioned research showed that the sensors’ operating frequency range depend on the geometric dimension, design technique, host medium, feeding network, etc. This research work proposes that metamaterials may be used to create multi-frequency based structures with smaller dimensions than conventional sensors because they support symmetric pairs with a unit cell structure and a negative effective properties at the resonant frequency. The use of a metamaterial in conjunction with the microstrip results in a multiband frequency whose size is determined by the lowest frequency. The proposed metamaterial-loaded microstrip-based sensor shows multiple resonant frequencies with the split-ring resonator inducing a circulating current and establishing the energy across the gap to produce the resonance in the different frequency regions. The addition of the dual-square split rings outside the area of the antenna increases the compactness of the design and enhances the sensitive area at the edge of the rings. The gain and realized gain reveal the amount of loss part in the design, which shows that the first and second frequencies provide less difference between this factor. Moreover, the linear correlation and R-square show a good agreement of the sensing performance for different sample materials. The compactness ratio for this proposed sensor is increased from 1 to 3.57 for the addition of the split rings in the microstrip design. The proposed metamaterial-based sensor increases the compatibility for noninvasive label-free detection and the compactness of the design.

## 7. Conclusions

The proposed metamaterial-based microwave sensor is designed for material characterization using a single-port-transmission-line-incorporated microstrip with dual-square split rings. A different MUT is placed on the sensor for monitoring the frequency deviation due to the variation in the permittivity of samples. This sensor has displayed the simplicity of the design where the metamaterial split rings increase the compactness of the design. The electric, magnetic, and surface current distribution exhibits the enhanced accumulation of electromagnetic spectrum on the surface of the structure and shows the high sensitive area for the proposed sensor. The sensitivity for different MUTs shows noticeable performance for two resonant frequencies of the reflection coefficient. A linear relation with the frequency deviation of the reflection coefficient is exhibited for different MUTs, where the R-squared is calculated near unity. This sensor can be applied for quality control processes and noninvasive characterization techniques for dielectric materials.

## Figures and Tables

**Figure 1 materials-15-07688-f001:**
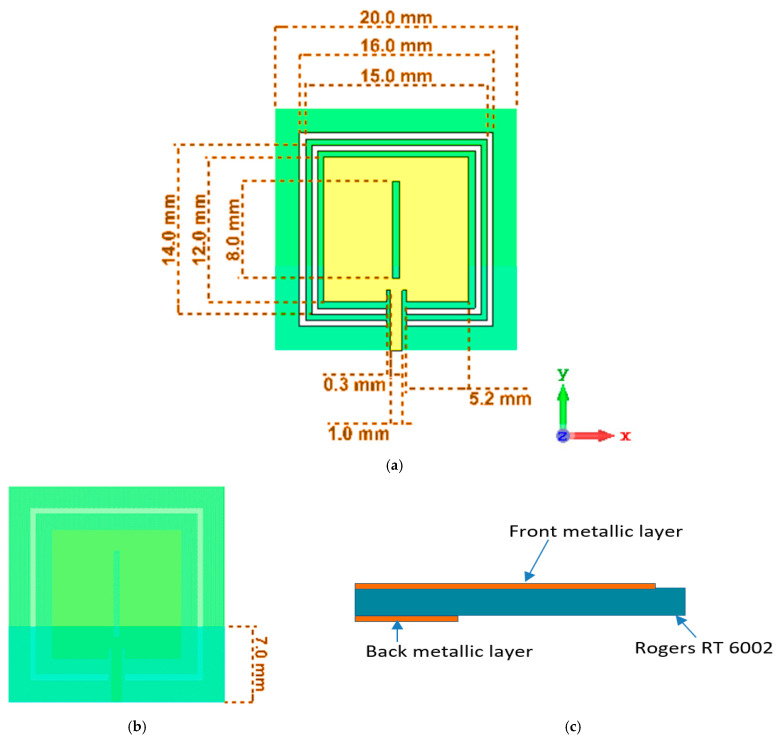
(**a**) Geometric configuration, (**b**) back view, and (**c**) side view of the microstrip-based sensor.

**Figure 2 materials-15-07688-f002:**
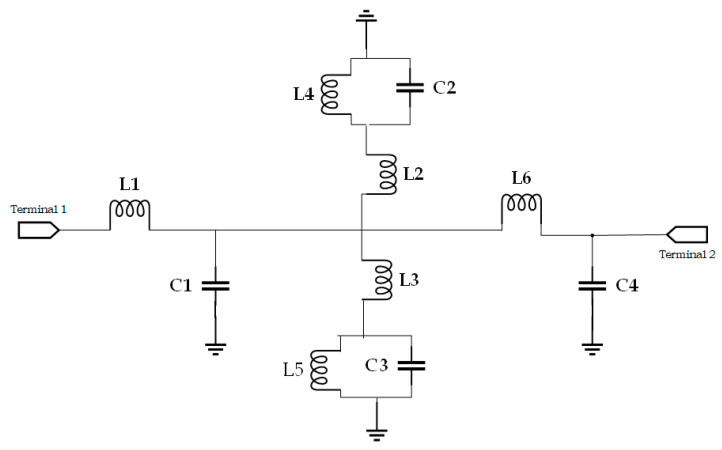
Equivalent circuit model for proposed metamaterial-based sensor.

**Figure 3 materials-15-07688-f003:**
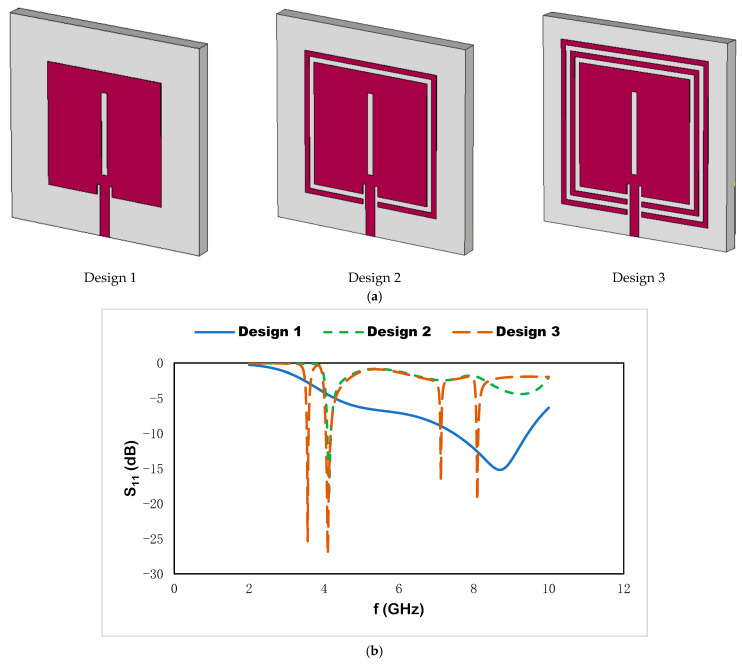
(**a**) Design steps and (**b**) reflection coefficient.

**Figure 4 materials-15-07688-f004:**
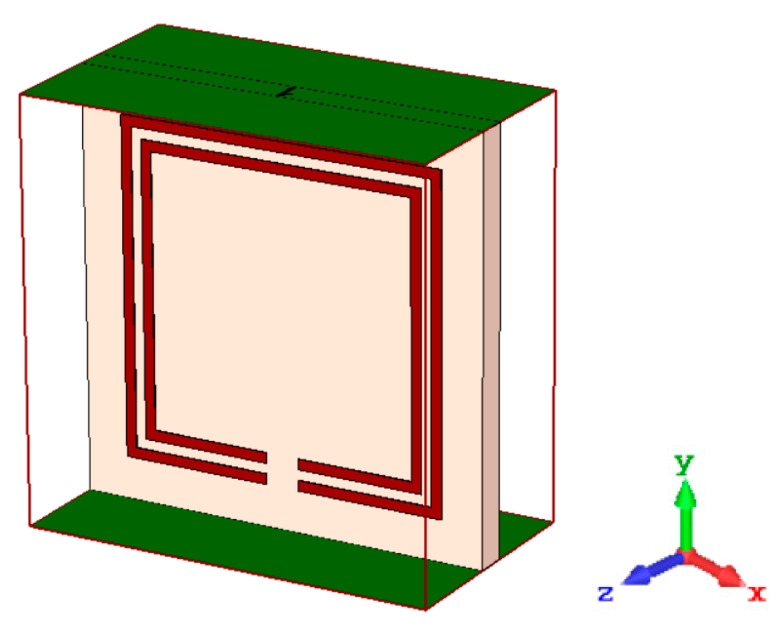
Boundary condition for metamaterial unit cell.

**Figure 5 materials-15-07688-f005:**
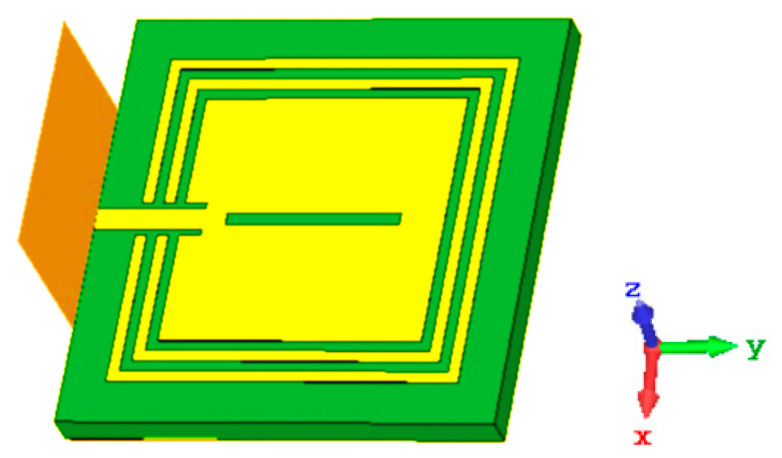
Single-port-feed-microstrip-based sensor.

**Figure 6 materials-15-07688-f006:**
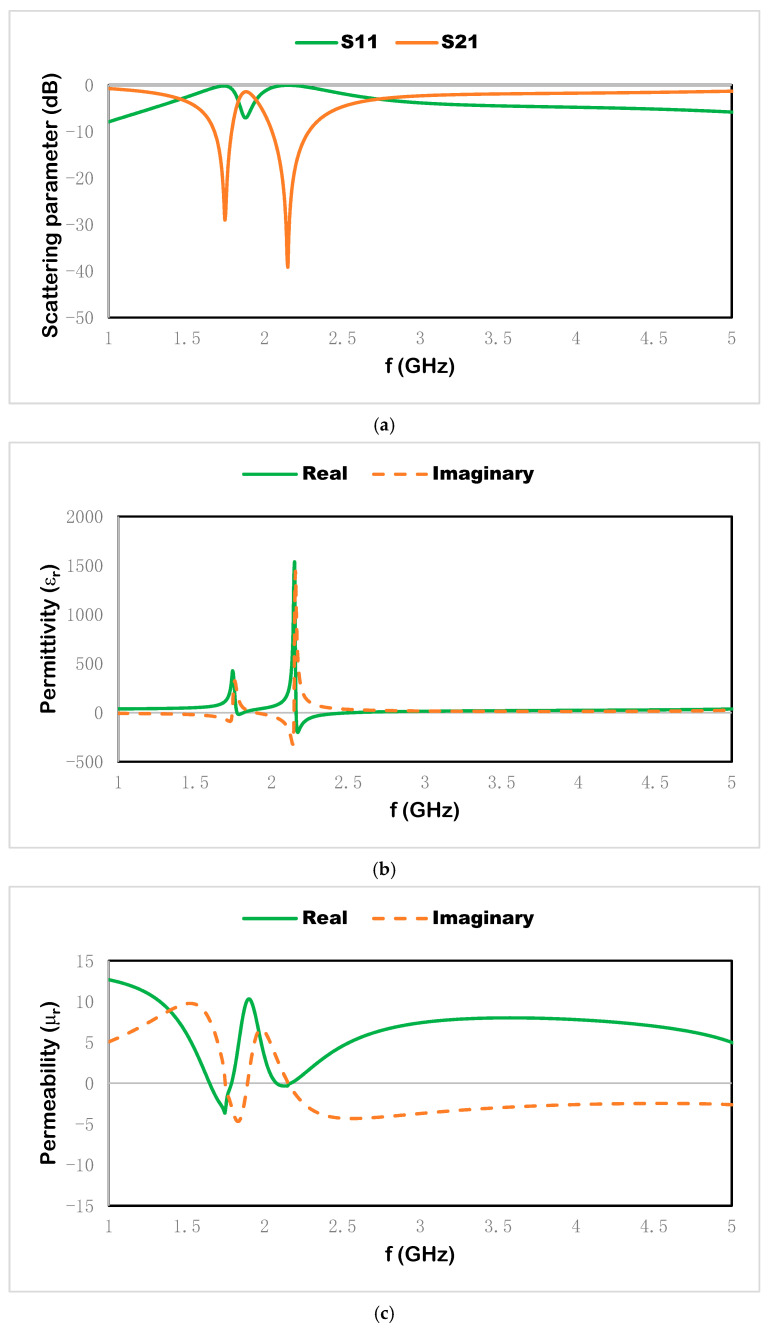
(**a**) Scattering properties, (**b**) effective permittivity, and (**c**) effective permeability.

**Figure 7 materials-15-07688-f007:**
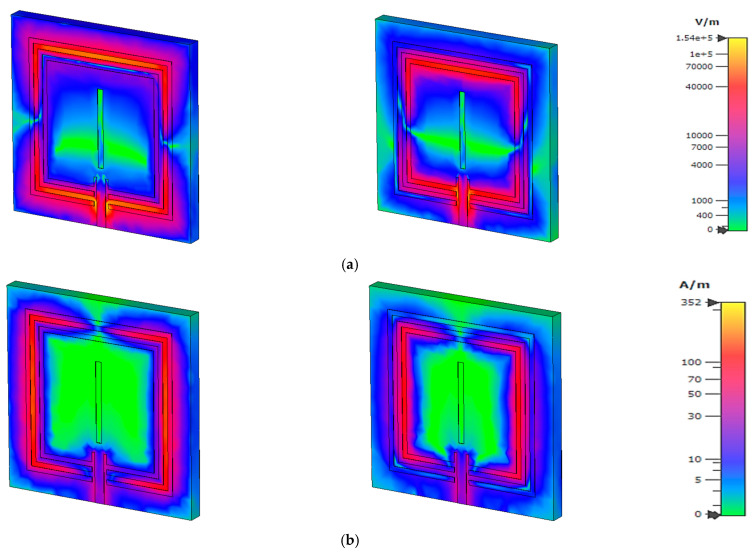
(**a**) Electric and (**b**) magnetic field, and (**c**) surface distribution at 3.56 and 4.14 GHz.

**Figure 8 materials-15-07688-f008:**
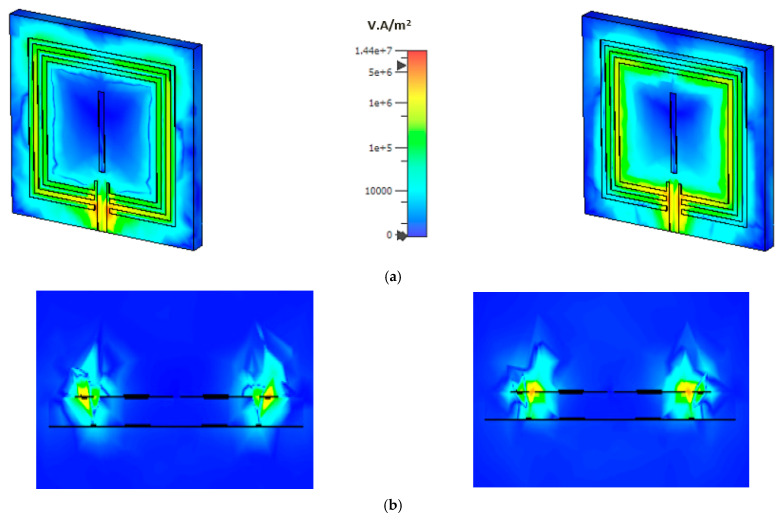
(**a**) Aerial view and (**b**) lateral view of power flow distribution on the left side at 3.56 GHz and right side at 4.14 GHz.

**Figure 9 materials-15-07688-f009:**
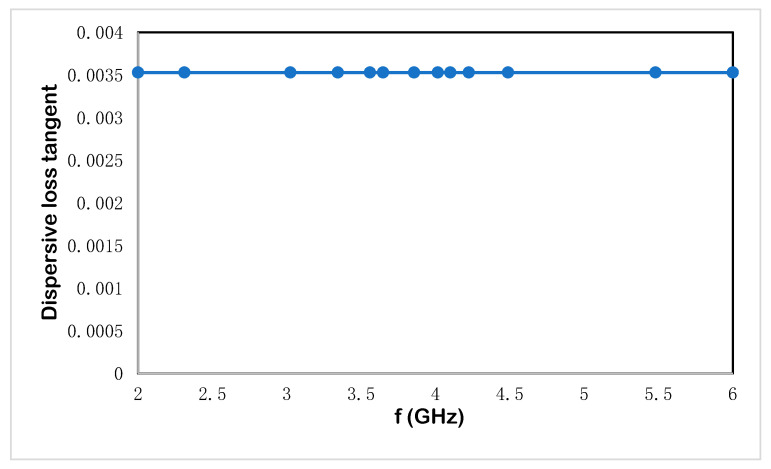
Dispersive loss tangent for the proposed split ring enclosed microwave sensor.

**Figure 10 materials-15-07688-f010:**
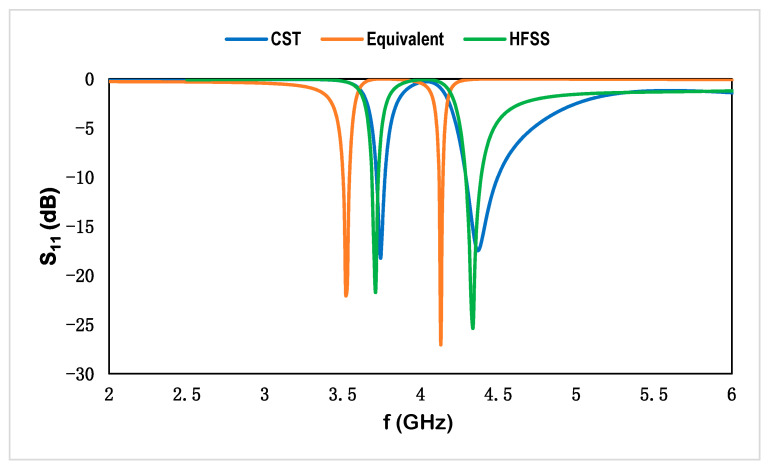
Reflection coefficient using different simulators for the proposed sensor.

**Figure 11 materials-15-07688-f011:**
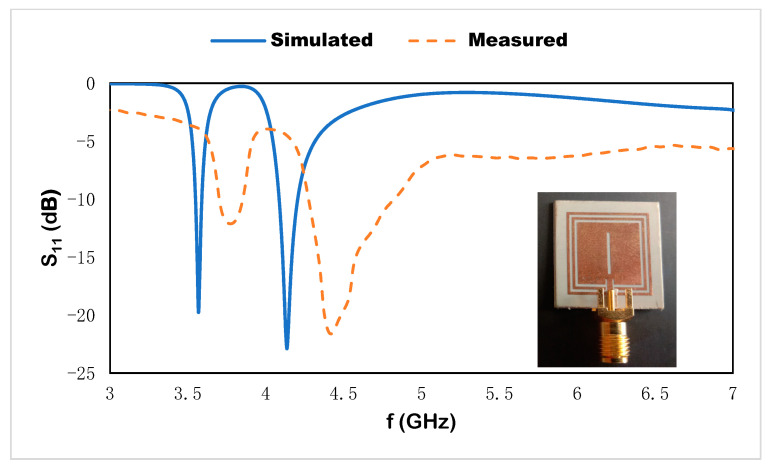
Simulated result and measured results of the reflection coefficient.

**Figure 12 materials-15-07688-f012:**
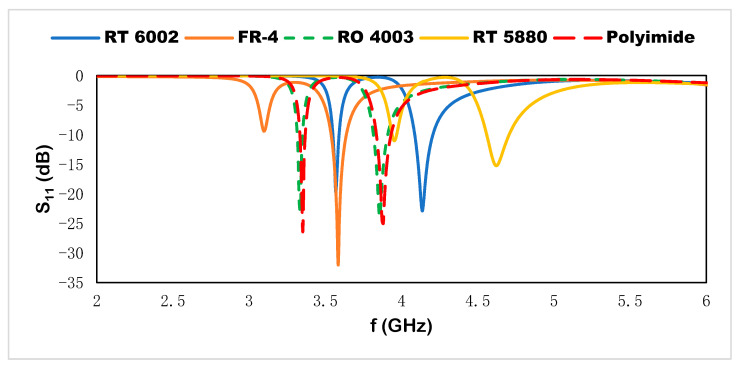
Different substrates as a host medium of the sensor.

**Figure 13 materials-15-07688-f013:**
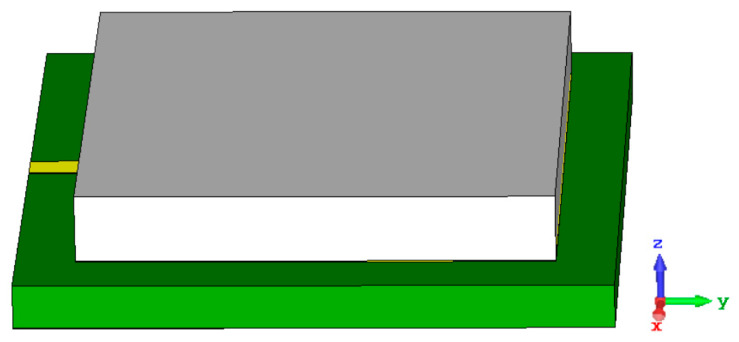
3D view of a proposed sensor with the MUT.

**Figure 14 materials-15-07688-f014:**
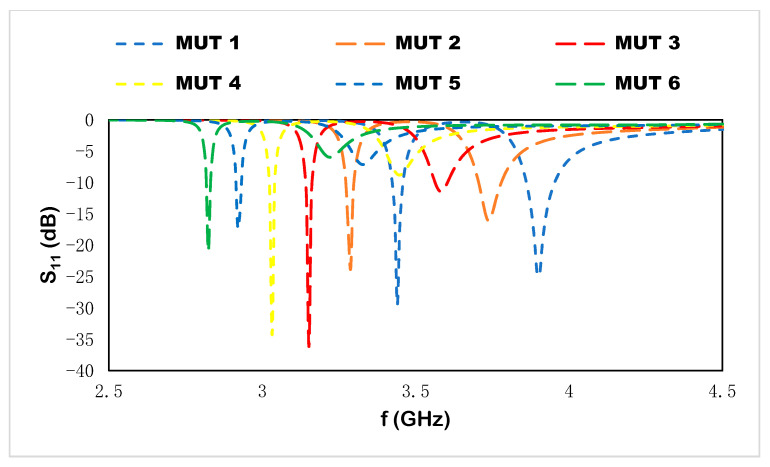
Reflection coefficient for different dielectric constants.

**Figure 15 materials-15-07688-f015:**
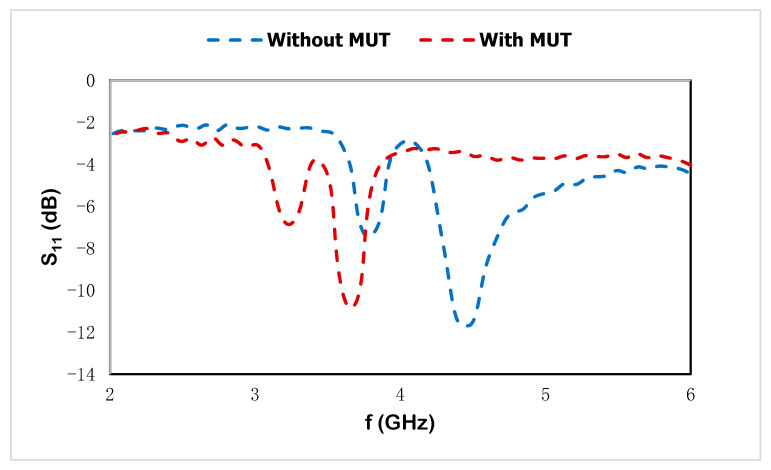
Measured results with MUT and without MUT.

**Table 1 materials-15-07688-t001:** The parametric specification for the proposed microstrip-based sensor.

Parameter	Length of the Outer Split Ring	Length of the Inner Split Ring	Width of the Outer Split Ring	Width of the Inner Split Ring	Cut Width in Microstrip Patch	Cut Depth of the Patch	Length of the Patch	Width of the Feed Line
**mm**	16	14	0.5	0.5	5.2	0.3	12	1

**Table 2 materials-15-07688-t002:** Different radiation properties for the proposed sensor.

	Resonant Frequency at 3.56 GHz	Resonant Frequency at 4.14 GHz
Radiation efficiency	−1.224	−0.3826
Total radiation efficiency	−1.270	−0.4056
Gain	2.092	2.819
Realized gain	2.047	2.797
Radiation plot	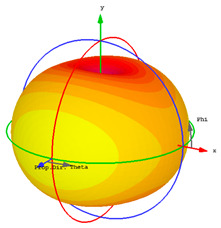	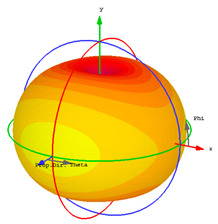

**Table 3 materials-15-07688-t003:** Dielectric constant and resonant frequencies with depth notch for the different substrates.

	Dielectric Constant	Resonant Frequencies	Depth Notch
FR-4	4.3	3.11, 3.6	−9.4, −32
Rogers RT 5880	2.2	3.94, 4.6	−11.5, −15.3
Rogers RO 4003	3.55	3.33, 3.84	−22.84, −24.24
Polyimide	3.5	3.4, 3.88	−26.32, −25.10
Rogers RT 6002	2.94	3.56, 4.14	−19.8, −23

**Table 4 materials-15-07688-t004:** Sensitivity performance for different dielectric constants.

	First Resonant Peak	Second Resonant Peak	Sensitivity
*ε_r_* = 1	3.56	4.14	-
*ε_r_* = 1.5	3.44	3.91	235
*ε_r_* = 2	3.29	3.74	450
*ε_r_* = 2.5	3.14	3.58	660
*ε_r_* = 3	3.03	3.46	860
*ε_r_* = 3.5	2.92	3.32	1000
*ε_r_* = 4	2.83	3.22	1170

**Table 5 materials-15-07688-t005:** Linear regression graph and regression equation at 3.56 and 4.14 GHz.

Linear Regression Plot	Linear Regression Equation	R^2^
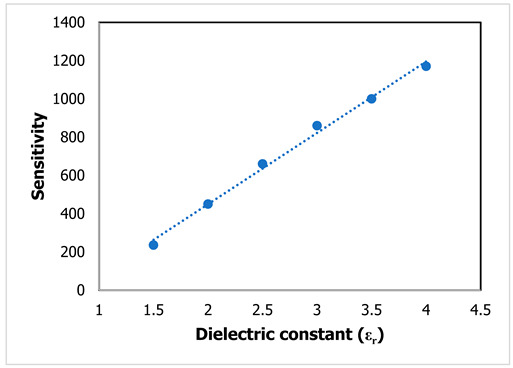	y = 372.86x − 296.19	R^2^ = 0.9943

**Table 6 materials-15-07688-t006:** Comparative study with different microstrip-based sensors.

	[20]	[21]	[22]	[23]	Proposed
**Substrate**	Rogers RO 4003	RO 4350	Rogers 5880	Quartz	Rogers RT 6002
**Type**	Pixelated CSRR based microstrip	CSRR based microstrip	Closed ring patch	Coplanar interdigital capacitor	Dual SRR enclosed microstrip
**Number of ports**	two	single	single	single	single
**Design simplicity**	medium	medium	simple	medium	simple
**Sensitivity parameter**	S21	S11	S11	Impedance	S11
**Dimension (mm^2^)**	50 × 50	30 × 25	38.69 × 68.12	0.3 × 0.7	20 × 20
**Operating frequency range (GHz)**	3–6	2.45–2.55	3.6–4.0	20–24	2.7–8.4
**Number of resonance frequency**	1	1	1	1	2
**Compactness ratio (λ/D)**	1.07	3.07	1.03	1.7	3.57
**Year**	2019	2020	2018	2020	2022

## Data Availability

All the data are available within the manuscript.

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
