# Peer review of "Dual-Square-Split-Ring-Enclosed Microstrip-Based Sensor for Noninvasive Label-Free Detection"

_materials, 2022, doi:10.3390/ma15217688_

Round 1
Reviewer 1 Report
Dear authors This work presents a refractive index sensing platform by introducing a dual squared-SRR resonator encircling a microstrip patch in the microwave range. First and foremost, the sensing mechanism and the dual-stacking resonators lack impressive innovation. The equivalent circuitry theory and EM simulation analysis are inconvincible in proving extraordinary performance among the counterparts of the proposed sensor. Moreover, the experimental characterization of the prototype is incomplete. Here below lists the points that need rectification and explanation. 1. The purpose of the joint analysis by unit-cell model and microstrip-line model is unclear. How did the dual-SRR change the reflection on the original microstrip-patched transmission line? The authors should give the resonant characteristics of the dual-SRR element and point out its influence on the sensing performance of the microstrip-line sensor. 2. Besides giving surface EM field and current distributions of the dual-SRR, the authors should provide aerial and lateral views of the power flux to show the field confinement for wave-matter interaction. According to the near-field distribution, the sensing area can be parametrically optimized and fully utilized. 3. Most of the figures are away from the publication standard. The corresponding figures should label all the symbols used in the context. Every subgraph should be numbered and given a subtitle. 4. Why is there noticeable deviance in terms of resonant frequencies and Q-factors by CST, HFSS, and equivalent circuitry in Fig.7? 5. I am confused about why the authors only presented the measurement result without sensing analytes. As observed in Fig.8, the deteriorated reflection coefficient may not beget expectant sensing performance. 6. According to the definition of sensitivity in Eq.(9), one can not evaluate the sensing performance in different frequency ranges. Thus, the sensitivity should be normalized by the referenced resonant frequency. In addition, I can not find substantial improvement in the proposed sensor in Table 6. 7. Similar work can be found in the previous publication of the first author. “Body-Centered double-Square Split-Ring Enclosed Nested Meander-Line-Shaped Metamaterial-Loaded Microstrip-Based Resonator for Sensing applications,” Materials 2022, 15, 6186. The authors should clarify the differences and what improvement the new design gets.
8. The English written in this manuscript is insufficient for publication. There are a lot of typos and grammar mistakes. For instance, the second sentence in the abstract, “The proposed sensor is consist of...” should be “The proposed sensor consists of....”; in Line 87, Page 2, the “polarisation” should be “polarization,” etc. Also, the usage of definite and indefinite articles is poor.
Overall, I do not recommend publishing this work based on these points.
Author Response
As attached.

Reviewer 2 Report
The paper “Dual square split ring enclosed metamaterial microstrip based 2 sensor for non-invasive label free detection” is interesting original paper is devoted to microwave sensor for material characterization based on metamaterial. The paper will be useful and interesting for researchers and should be published after corrections.
1. The authors do not write whether they are going to use in thickness only one or several cells, and do not give the thickness of the cell, neither in the text, nor in figures 1-3. Therefore, the question arises - whether this structure of metamaterial can be considered precisely as a metamaterial or as a metasurface. In the case, when the structure should be metasurface, another consideration is necessary.
2. It is not written where formula 1 came from.
3. In many places in the article, the internal logic of the presentation of the study is not clear.
So, at the lines 176-178 it is written that “Fig. 5 the effective parameters such as effective permittivity, and effective permeability 176 ability are presented for the metamaterial unit cell. The real negative permittivity value is 177 exhibited from 5.32-5.42, 6.57-6.64, 7.22-7.82 GHz and real negative permeability is shown 178 from 6.5-6.6, 7.1-7.5 GHz.”
Then authors proceed to present the electric field distribution at 3.56 GHz. Why the frequency 3.56 GHz is chosen? And there are no further comments and studies of electric and magnetic field distribution at frequencies where there are negative values of dielectric permittivity and magnetic permeability? For example, it was possible to compare these values and distributions in regions with positive and negative values of dielectric permittivity and magnetic permeability.
4. In the line 281, it is written “For ? =1.5, the scattering properties are changed due to the introduction of the MUT.” Apparently, it is not ?, but ?r.
5. The same in the line 289 and below the frase «the consequently, the rest of samples with variation in the dielectric constant to 4, shows 289 the sensitivity performance for two resonant frequencies (76, 96) for ? = 2, (117, 135) for ? 290 = 2.5, (148, 164) for ? = 3, (179, 198) for ? = 3.5 and (205, 222) for ? =4 respectively.» Apparently, it is not ?, but ?r. According to table 4 above, there are no such ?, but there is such ?r. dielectric constants.
6. Table 3. „Dielectric constant, resonant frequencies with depth notch for the different substrates”. The dielectric constant is the dielectric permittivity at zero frequency. In the paper high frequency values are used. In this case, it should be shown the estimation the frequency dispersion of materials.
Author Response
As attached.

Reviewer 3 Report
Dual square split ring enclosed metamaterial microstrip based 2 sensors for non-invasive label-free detection Air Mohammad Siddiky et al.
Manuscript materials-1923305
1. The title is not appropriate: 1) the report consists of the theory, simulation, optimization, and fabrication of the sensor rather than focusing on its application non-invasive label-free detection
2) This component is not metamaterial. like your reference [18-21], they do not call their structure metamaterial as you show in Table 6. Also, it is only a single unit as shown in the inserted photo of your fabricated prototype in figure 8.
2. The paper can be shortened. For example, for the introduction, you can mention the examples only related to your scope in a logical way. I suggest also including fewer figures and fewer tables.
3. If you would like to simulate a metasurface, the ZY-plane and ZX-plane should be set as periodical boundary conditions.
4. Page 14, Table 6 row 7, the dimensions should have a unit.
Overall, I think the authors did lots of work on this topic, I think the simulation for different substrates and the fabricating of the prototype are the two highlights of this paper. I suggest just cleaning up and reorganizing the draft a bit, we can consider accepting to publish.
Very good!
Author Response
As attached.

Round 2
Reviewer 1 Report
Second round
The revision still lacks sound evidence to prove its novelty and performance.
1. The logic is out of order. What is the relation between Fig.3(b) and Fig.6(a)? Introducing metamaterial units into the microstrip sensor utilizes the negative permittivity and permeability to enhance field confinement. However, Fig.6(a) shows different frequency ranges of negative properties with Fig.3(b).
2. I still doubt the simulation difference between HFSS and CST. It can not be that significant.
3. The sensitivity expression in Equation 9 does not make sense to me. Please refer to their previous publication on Materials and explain why the sensitivity definition changed.
Author Response
As attached.

Round 3
Reviewer 1 Report
I think the revision has improved the quality of the manuscript. Notwithstanding some minor problems, I agree to accept it for publication.